# Exploring physician approaches to conflict resolution in end-of-life decisions in the adult intensive care unit: protocol for a systematic review of qualitative research

Harleen Kaur Johal ⬡ , Giles Birchley, Richard Huxtable

Centre for Ethics in Medicine, Bristol Medical School, University of Bristol, Bristol, UK

**Correspondence to**
Dr Harleen Kaur Johal;
harleen.johal@bristol.ac.uk

## ABSTRACT

**Introduction** Conflict is unfortunately well-documented in the adult intensive care unit (AICU). In the context of end-of-life (EOL) decision-making (ie, the withdrawal or withholding of life-sustaining treatment), conflict commonly occurs when a consensus cannot be reached between the healthcare team and the patient's family on the 'best interests' of the critically ill, incapacitated patient. While existing literature has identified potential methods for conflict resolution, it is less clear how these approaches are perceived and used by stakeholders in the EOL decision-making process. We aim to explore this by systematically reviewing and synthesising the published evidence, which addresses the following research question: what does existing qualitative research reveal about physician approaches to addressing conflict arising in EOL decisions in the AICU?

**Methods and analysis** Peer-reviewed qualitative studies (retrieved from MEDLINE, Project Muse, Scopus, EMBASE, Web of Science, PsycINFO, CINAHL, and LILACS) examining conflict and dispute resolution in the context of EOL decisions in the AICU setting will be included. Two reviewers will independently screen either all or a randomly selected sample of studies, with a third reviewer independently screening studies of uncertain eligibility. The 'thematic synthesis' approach will be employed to analyse the resulting data. The quality of included papers will be assessed using the 2018 Mixed-Methods Assessment Tool. The 'Grading of Recommendations, Assessment, Development, and Evaluations-Confidence in the Evidence from Reviews of Qualitative research' approach will be used to assess our confidence in the findings.

**Ethics and dissemination** Ethical approval is not required for this review, as only published data will be included. We anticipate that the findings will be of interest to healthcare professionals working in AICUs and individuals working in bioethics, given the ethically contentious nature of EOL decisions. The findings will be disseminated at academic conferences and through open-access publication in a peer-reviewed journal.

**PROSPERO registration number** CRD42021193769.

## INTRODUCTION

Conflict is unfortunately well-documented in the adult intensive care unit (AICU). In the

## STRENGTHS AND LIMITATIONS OF THIS STUDY

⇒ This systematic review has clear scope, with pre-defined inclusion and exclusion criteria.
⇒ The reviewers will identify additional articles of interest, by hand-searching the reference and citation lists of included papers.
⇒ The search strategy benefits from using a range of bibliographic databases, including those which index both philosophical and clinical research.
⇒ The exclusion of other healthcare professionals' approaches (such as nurses') to resolving conflict may result in effective approaches not being identified.
⇒ By focusing on qualitative data, rather than theoretical or normative research, the findings of this review will better represent the reality and feasibility of conflict resolution in clinical practice.

context of end-of-life (EOL) decision-making (ie, the withdrawal or withholding of life-sustaining treatment), it commonly occurs when a consensus cannot be reached on the 'best interests' of the critically ill, incapacitated patient.[1] Families, as surrogate decision-makers, may favour preservation of life, while physicians may be reluctant to provide life-sustaining treatment, which is potentially inappropriate.[2] The decision-making process for healthcare professionals (HCPs) is informed by their understanding of the law and professional guidance. In England and Wales, when there is an agreement on the best interests of a patient, there is no need to make an application to the court to withdraw or withhold life-sustaining treatment.[3] Involvement of the courts is however necessitated where disagreement between the conflicting parties has become intractable. This adversarial process exacerbates distress among HCPs and families, potentially leading to breakdowns in therapeutic relationships.[4] This arguably justifies the utilisation

of alternative methods of conflict resolution, before a dispute becomes sufficiently entrenched to require legal intervention.[1 5]

There is an extensive body of literature recognising the incidence of conflict in adult ICUs. The 'Conflicus' study surveyed the experiences of 7498 ICU staff members in 323 ICUs in 24 countries. Nurse-physician conflicts were most common (32.6%) and staff-relative conflicts accounted for 26.2% of perceived conflicts. Lack of psychological support and problems with the decision-making process were identified as causes of conflict in EOL care.[6] Indeed, in a critical literature review exploring doctors' and nurses' EOL decision-making process, Flannery et al suggested that more comprehensive and standardised approaches are needed to support all HCPs in making these difficult decisions.[7] One could extend this suggestion into a need for more guidance, to support HCPs in the resolution of disputes when they occur. While academic literature has identified potential routes for dispute resolution, for example, clinical ethics committees and mediation,[8] it is less clear how these approaches are perceived and used by stakeholders in the EOL decision-making process, such as HCPs, patients and surrogate decision-makers.[9] It has also been observed in other areas of healthcare that theoretical discussion of ethical challenges is not necessarily representative of the ethical issues that arise in real clinical practice.[10] Further consideration should therefore be given to how these stakeholders actually approach conflict and how they perceive strategies for conflict resolution.

We seek to explore this by systematically reviewing and synthesising the published qualitative evidence on physician approaches to conflict resolution in EOL decisions in the AICU. A coherent and cohesive understanding of current approaches to conflict resolution will provide a starting point for an evidence-base, from which HCPs can be trained in how to approach conflict.[11] The rationale for this review is further supported by recent developments in the field of bioethics, where the value of empirical evidence in evaluating normative claims has been increasingly recognised.[12] Hence, in order to address the question of how conflict should be resolved, we must first consider the context of conflict and how physicians attempt to resolve it, so the resulting ethical analysis is informed by and more relevant to clinical practice.[13]

This systematic review shall lay the foundations of a larger empirical bioethics study, in which disagreements between HCPs and patient representatives will be explored. Ultimately, the development of guidelines to support all HCPs, patients, and families when conflict arises, will aim to improve their experience during an ethically contentious and emotionally strenuous time.[14]

## Aims

This systematic review protocol has been guided by the Preferred Reporting Items for Systematic Reviews and Meta-Analyses Protocols (PRISMA-P) checklist,[15] and the review will aim to address the following research question: what does existing qualitative research reveal about physician approaches to addressing conflict arising in EOL decisions (specifically, the withdrawal and withholding of life-sustaining treatments) in the AICU? The standard 'Participants, Interventions, Comparators, and Outcomes' system used in reviews of clinical studies has been modified here, as it is less suited to qualitative evidence synthesis, by using the 'Methodology, Issue, Participants' (MIP) system by Strech et al.[16]

## METHODS AND ANALYSIS
### Eligibility criteria
The purpose of this review is to identify studies in which physicians have described their strategies for approaching and resolving conflict around EOL decisions in the AICU. The inclusion and exclusion criteria, following the MIP system by Strech et al, are shown in table 1. Peer-reviewed qualitative studies examining conflict resolution in the context of EOL decisions in the adult intensive care setting will therefore be included.

Studies identifying factors, which contribute to the development of conflict in AICU around EOL decisions, will not be included. Identification of these factors is important in the development of conflict resolution strategies, however this would constitute a separate systematic review. Studies relating to conflict around EOL decisions for critically ill children and neonates will also be excluded. While there may be similar circumstances to those which present in an adult intensive care environment, where the patient is incapacitated and a decision must therefore be made in their best interests, there are legal and clinical differences between adults and children. The role of the 'family and others' is notably different in best interests decision-making processes for adults,[17] as compared with children; and the resulting conflicts may be different in nature. While conflict resolution strategies in AICU could be applied in paediatric ICU and vice versa, it is beyond the scope of this review to consider two clinical contexts, with potentially crucial differences, and two groups of stakeholders with differing roles.

### Search strategy
Scoping searches were conducted, using MEDLINE Subject Headings (MeSH) and truncations of keywords obtained from a previous systematic review of palliative care in the AICU.[18] The search strategy was further refined following consultation with a librarian, by checking the MeSH database to identify relevant concepts and select appropriate terms (see Author notes). Descriptors and synonyms were both used, in addition to truncations and abbreviations of terms, to retrieve all related variants.[19] The MEDLINE search strategy was then amended for other databases as needed, including adjustments to subject headings (see online supplemental file 1).

Scoping searches for the study began in March 2021, and the planned end date for the systematic review is July 2022.

**Table 1**  Inclusion and exclusion criteria

|  | Inclusion criteria | Exclusion criteria |
|---|---|---|
| Context | Critical/intensive care settings where EOL decisions are made for adults will be included.<br>The terms 'critical' and 'intensive' are often used interchangeably for clinical settings in which patients are seriously ill and/or require some form of life-sustaining treatment. | Clinical settings, in which EOL decision are made for children or neonates will be excluded.<br>Non-critical/intensive care settings will be excluded.<br>Other specialities or settings, in which EOL decisions are made (eg, community palliative care), will be excluded. |
| Methodologies | Qualitative studies examining conflict resolution in the AICU (eg, interviews and focus groups).<br>Mixed-methods studies (eg, surveys with open questions) will also be included for consideration of the qualitative evidence presented. | Non-empirical studies examining conflict resolution. This may include normative and theoretical literature. These have been excluded as we are interested in understanding the lived reality of conflict resolution in clinical practice.<br>Quantitative studies examining conflict resolution in the AICU. These have been excluded as we are interested in deeper exploration of experiences in the AICU, which is not possible using quantitative methods. |
| Issues | Qualitative studies exploring conflict resolution strategies within the AICU, around EOL decisions, will be included.<br>EOL decisions are here defined as decisions relating to the withdrawal or withholding of life-sustaining treatment (eg, dialysis, clinically assisted nutrition and hydration, and artificial ventilation), or decisions relating to cardiopulmonary resuscitation.<br>Conflict is here defined as a failure to reach an agreement on whether life-sustaining treatment should be withdrawn or withheld. The terms 'conflict', 'dispute', 'disagreement', 'dissent', and 'refusal' are often used interchangeably in the existing literature, and we are therefore interested in studies which explore any of these issues. | Literature that does not explore conflict, dispute, disagreement, dissent, and refusal will be excluded.<br>Literature that does not explore withdrawal or withholding of life-sustaining treatment will be excluded.<br>Empirical studies examining factors, which contribute to conflict, will be excluded. While the identification of factors which cause conflict is fundamental to the development of conflict resolution strategies, it is beyond the scope of this systematic review. |
| Participants | Studies which explore physician approaches to conflict resolution will be included.<br>Studies which explore other stakeholders' perceptions of physician approaches to conflict resolution will also be included. Stakeholders are broadly defined in three categories: (i) healthcare professionals (eg, physicians/doctors, nurses, and therapists), (ii) adult patients, and (iii) patient representatives (eg, families, spouses, relatives, and other surrogate decision-makers). | Literature that does not encompass discussion of stakeholders' (as defined in the inclusion criteria) approaches to conflict in EOL decisions in the AICU, will be excluded.<br>Literature that discusses non-physician-led approaches to conflict resolution, for example, clinical ethics consultation or independent mediation, will be excluded. |
| Timeframe | Any studies published after 2000 will be included, as critical care is a rapidly developing field. | Studies published before 2000 will be excluded, as they are less likely to be relevant to current clinical practice. |
| Types of publications | Peer-reviewed journal publications of empirical research.<br>Publications in English.<br>International publications will be included. | Unpublished and grey literature, theses, and dissertations, and any published sources that do not contain empirical studies will be excluded.<br>Publications not in English will be excluded.<br>Conference abstracts will be excluded.<br>Review articles relating to the research question will be used for the identification of empirical studies only.<br>Study authors will be contacted if we are unable to obtain the full text through the university subscription. If the study authors do not respond to this request within two weeks, the study will be excluded. |

AICU, adult intensive care unit; EOL, end-of-life.



## Information sources

We will use a combination of general bibliographic (Web of Science, Scopus), subject-specific bibliographic (MEDLINE, EMBASE, PsycINFO, CINAHL, and LILACS), and full-text (Project MUSE) databases. These databases have been chosen as they are known to be directly relevant to studies conducted in medicine, social sciences, and bioethics. We have imposed a restriction on publication dates, as critical care is a rapidly developing field and studies pre-2000 are likely to be outdated. We will also only include papers in English. Due to the resource-intensive nature of reviewing grey literature, it will also be excluded.[20]

The reference and citation lists of papers found to meet inclusion criteria will be scrutinised for further relevant papers. In addition, the corresponding authors will be contacted for full-text versions of relevant abstracts, which are not available through the University subscription. Where authors have not responded within two weeks, the study will be excluded.

## Selection process

The initial screening of all the titles/abstracts of the full search results, to determine whether papers meet the inclusion criteria, will be performed by the first reviewer (HKJ). The full search results will be sorted into one of three categories: 'include', 'exclude', and 'unsure'. Any issues in screening at this point will be discussed with the research team, to further develop the inclusion/exclusion criteria. The second reviewer (William Orchard) will independently screen a randomly selected 10% of each of the included and excluded papers, and all of the 'unsure' papers. Any remaining contentious titles/abstracts will be re-assessed and independently screened by a third reviewer (GB).

All full-text versions of relevant titles/abstracts will then be retrieved and evaluated for eligibility by the first reviewer (HKJ). Where there is uncertainty around the eligibility of any papers, these will be discussed between two reviewers (HKJ and WO). If the disagreement persists, a third reviewer (GB) will make a final decision.

## Data extraction and management

Search results will be exported into EndNote X9, where the data will be stored and managed. Duplicate results will be removed and numerical results of each stage of the systematic review will be illustrated in a PRISMA flow diagram.[21]

Data will be extracted by the first reviewer (HKJ) from a sample of the included studies, using a preliminary form. The data extraction form will then be revised as needed, following discussion with the research team, and the first reviewer (HKJ) will extract data from all included studies using the pilot-tested extraction form. The four proposed domains in the data extraction form are: (i) reference details (title, publication year, authors, journal); (ii) study details (aims, study setting, methods, participant characteristics); (iii) results and key findings; (iv) limitations (eg,

evidence of bias). If data are missing, the study authors will be contacted. Where authors have not responded within two weeks, the data will be excluded.

The second reviewer (WO) will independently extract data from 10% of the included studies. These data extraction forms will be compared between the first two reviewers, to assess for inter-rater reliability and ensure homogeneity.

## Risk of bias (quality) assessment

The 2018 Mixed-Methods Assessment Tool (MMAT) will be used to assess the quality of individual studies, as it permits the appraisal of qualitative studies and mixed-methods studies,[22] both of which are in our inclusion criteria. The quality of each included study will be assessed independently by two reviewers. As quality assessment of qualitative sources is notoriously subjective,[23] studies of low methodological quality will still be included. We will however discuss low scores and their reasons in the final reporting, if a low-quality paper makes significant and unique contributions.

## Data synthesis

Given that this review focuses predominantly on qualitative research findings, traditional methods of data synthesis found in aggregative reviews, such as meta-analysis, will not be appropriate. To account for and standardise data synthesis in systematic reviews which include a range of research designs, the UK ESRC Methods Programme has produced guidance on conducting 'narrative synthesis'.[24] A variation of this framework, adapted by Schofield et al,[25] to synthesise data not focused on a particular intervention, will be used. A preliminary synthesis of the data, using the methods of 'thematic synthesis' developed by Thomas and Harden, will be undertaken to integrate themes and content that emerge from the included studies. Thematic synthesis has three stages: the coding of text 'line-by-line', the development of 'descriptive themes', and the generation of 'analytical themes'.[26] NVivo software will be used to undertake this qualitative data analysis. Relationships in the data will subsequently be explored, within and across the included studies. The strength of the evidence will then be assessed and the data will finally be synthesised to develop a theoretical model of physician approaches to conflict resolution in the AICU.[24]

As we aim to synthesise qualitative data, the 'Grading of Recommendations, Assessment, Development, and Evaluations-Confidence in the Evidence from Reviews of Qualitative research' (GRADE-CERQual) approach has been chosen to assess confidence in the findings of this review, as it provides a systematic and transparent framework for assessing confidence in qualitative evidence synthesis. This is based on the consideration of four components: (i) methodological limitations, (ii) coherence, (iii) adequacy of data, and (iv) relevance. This will permit an assessment of whether the review findings are a reasonable representation of the phenomenon of interest.[27]

## Patient and public involvement

There was no patient or public involvement in designing this protocol, although the findings will inform planned research with patients and HCPs.

## Ethics and dissemination

Ethical approval is not required for this review, as only published data will be included. We anticipate that the findings will be of interest to HCPs working in ICUs and individuals working in conflict resolution spheres (eg, lawyers and mediators). The findings are also likely to be of interest to researchers in these fields, as well as those within the interdisciplinary field of bioethics, given the ethically contentious nature of EOL decisions. We aim to disseminate findings both at academic conferences and through open-access publication in a peer-reviewed journal.

## DISCUSSION

This systematic review will be the first, to our knowledge, to synthesise physician approaches to conflict resolution in EOL decisions in the AICU. While previous reviews have explored the experiences of EOL decision-making for ICU HCPs, or family satisfaction with EOL care in ICUs,[7 18] this will be the first to focus specifically on methods, employed by physicians, to resolve conflict. We hope that the findings of this review will therefore go on to inform educational curricula and training for ICU HCPs, who may face conflict in EOL decision-making. It is also feasible that the findings may be transferable to other medical specialties, in which EOL conflicts may also arise, such as geriatrics and palliative care. Furthermore, we hope the findings will benefit individuals working in conflict resolution circles, as the review will evaluate the existing evidence to highlight both helpful and unhelpful methods. Additionally, by establishing what is currently known about physician approaches to resolving conflict, gaps in the evidence base can be identified, in order to guide future research.

There are potential limitations to this review. First, focussing on EOL disagreements may limit the identification of conflict resolution methods that are successfully employed in AICUs, to resolve other forms of conflict. For example, conflicts over pain management would not be included in this study, but may contribute to discord between a patient's representatives and the responsible physician. However, given the intensity of EOL conflicts, if we are able to identify methods that physicians use to successfully resolve these conflicts through our review, these methods could potentially be employed to a certain degree in resolving other forms of conflicts (both within and outside the AICU). It is also a necessary restriction to make the review more feasible.

Second, we focus solely on physicians' approaches to resolving conflicts (while taking stakeholders' perceptions of physician approaches into account). This may seem to overlook the role of nursing staff, who spend the most time at the patient's bedside, and therefore develop close relationships with both the patient and relatives.[7] It has been found that in critical care, nurses develop a sense of advocacy—that is, an obligation to support the patient's best interests. This may not always align with the physician's view and arguably contributes to the high prevalence of nurse-physician conflicts.[6] The relationship between nurses and patients is certainly of interest, as nursing staff may also offer strategies to resolve conflicts. However, as Flannery *et al* reported, the responsibility for EOL decision-making ultimately lies with the physician, and we have therefore chosen to concentrate our review on physician approaches to resolving conflicts.[7] Moreover, this review does not consider external interventions, for example, legal advice, clinical ethics consultation, and mediation. Although these methods have their own place and importance within conflict resolution spheres, we are particularly interested in the strategies that physicians employ in their day-to-day practice within the AICU, to address EOL disagreements when they arise. Considering all methods of EOL dispute resolution is beyond the scope of this systematic review, however they will be considered in a narrative review that is also being undertaken as part of the larger empirical bioethics study.

Finally, we have chosen not to apply methodological filters to identify qualitative research, due to the concern that methodological filters may result in loss of relevant studies. Instead, keywords have been chosen to identify qualitative research (see online supplemental file 1). These keywords were chosen based on previous systematic reviews and pilot searches. They have also been adapted to the keyword catalogues and indexing vocabulary in each database. Following a pilot search on MEDLINE, the keywords retrieved the same studies as a qualitative research methodological filter, and more. Using these keywords successfully identified studies of interest, which were known to the research team prior to commencing the review. It is still possible that a relevant study may be missed by the search strategy. We hope to circumvent this, by hand-searching the reference lists and citation lists of included studies. Additionally, how to best assess the quality of qualitative studies is widely debated. Due to our inclusion of the qualitative data from mixed methods studies, using the MMAT will allow us to use one tool to assess the quality of both purely qualitative and mixed methods studies. The MMAT has fewer criteria than alternatives (eg, the Critical Appraisal Skills Programme Qualitative 2018 checklist[28]) and may therefore not provide as thorough an assessment of quality. However, as we do not intend to exclude studies of low methodological quality, the MMAT will still allow us to factor low scores into our overall analysis of the data.

**Acknowledgements** The authors would like to thank Sarah Herring, for her help in refining the search strategy for this review and navigating online databases; William Orchard, for his input as the second reviewer; and the reviewers of this manuscript, for their helpful suggestions.



**Contributors** HKJ, GB and RH conceived of the review, developed the protocol, and developed the search strategy. HKJ drafted the manuscript and is the guarantor of this review. All authors revised and edited the draft manuscript, and approved the final version.

**Funding** The work is based at the University of Bristol, as part of the 'Balancing Best Interests in Health Care, Ethics and Law (BABEL)' Collaborative Award. This research was funded in whole, or in part, by the Wellcome Trust Grant no. 209841/Z/17/Z. For the purpose of Open Access, the author has applied a CC BY public copyright licence to any Author Accepted Manuscript version arising from this submission.

**Disclaimer** The funders had no role in the preparation of this manuscript or the decision to submit for publication.

**Competing interests** None declared.

**Patient and public involvement** Patients and/or the public were not involved in the design, or conduct, or reporting, or dissemination plans of this research.

**Patient consent for publication** Not applicable.

**Provenance and peer review** Not commissioned; externally peer reviewed.

**Author note** While some of the subject heading terms may appear contradictory, we have included all terms that relate to the withdrawal or withholding of treatment, regardless of whether this is done as part of a research protocol (eg, MeSH term "Withholding Treatment"), or whether treatment is withheld by the professional against the patient's or patient's representatives' wishes (eg, MeSH term "Refusal to Treat"). This is to identify any potential conflicts that may arise when making decisions about life-sustaining treatment. Many thanks to Dr Dina Vrkić for highlighting this.

**ORCID iD**
Harleen Kaur Johal http://orcid.org/0000-0002-8665-8932

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
