## [Reviewer comments · BMJ Open]

ARTICLE DETAILS

TITLE (PROVISIONAL)	Protocol for a Systematic Review of Qualitative Research Exploring Physician Approaches to Conflict Resolution in End-of-Life Decisions in the Adult Intensive Care Unit
AUTHORS	Johal, Harleen; Birchley, Giles; Huxtable, Richard

VERSION 1 – REVIEW

REVIEWER	Vrkić, Dina University of Zagreb, Central Medical Library
REVIEW RETURNED	01-Nov-2021

GENERAL COMMENTS	The authors of the proposed protocol are off to a good start; however, this protocol requires additional clarification and possible changes in the search strategy and information sources. Although it is a methodological issue, it should not disturb the flow of the proposed protocol. In the search strategy, the authors are referring to the search strategy from the paper (ref. 18 - Schram A, Hougham G, Meltzer D, Ruhnke G. Palliative care in critical care settings: A systematic review of communication-based...) which is not so applicable for this protocol because it is off-topic. Therefore, the proposition is that the authors use mentioned article as an inspiration or starting point for key search terms in this protocol and explain why they used their selected MeSH terms and keywords. The primary issue is a potential concept issue within the search strategy. Definition of the EOL decisions and relating them to “the withdrawal or withholding of life-sustaining treatments” conflict with the search term “Refusal to Treat” and focus on the aim of this protocol. Therefore, the authors should clarify the following section to avoid confusion. Please bear in mind that the “withdraw*” and “withhold*” keywords can connect to the “Withholding Treatment” MeSH term (https://www.ncbi.nlm.nih.gov/mesh/68028761), which is opposite of the “Refusal to Treat” MeSH term (https://www.ncbi.nlm.nih.gov/mesh/68016079) as is defined as: “Refusal of the health professional to initiate or continue treatment of a patient or group of patients”. Due to the wide range of years in the search, the search strategy needs to be more flexible to gather possible terms. For example, not all papers are indexed under the MeSH term “Terminal Care” from 1980 – until today. Therefore, the recommendation is that the authors need to include all possible MeSH and all possible keywords to the related term(s). In this specific case, it could also be helpful to use the MeSH term “Terminal Ill”, “Palliative Care”, etc. To be sure that all (possible) papers are retrieved.
--

	One suggestion, for the keyword, search the .mp search field is excellent in the Medline database for solo search, but because there is a multiple database search, I advise authors to be consistent in the search and search fields across the other databases. Please consider to use .ab (abstract) and .ti (title) fields instead of the .mp. Also, please consider using the abbreviations of the EOL, (A)ICU in the search strategy because we never know how the title, abstract, or keywords are constructed. For example, authors can use (Ovid MEDLINE) for the “End of life”: (End adj2 life).ab,ti (End adj2 life care).ab,ti EOL.ab,ti. EOLC.ab,ti. EOL care.ab,ti. Suggestions are on the same page as the definition and application of the methodological filters that the authors mentioned in the discussion. The description of Boolean operators is redundant in describing the search strategy. (Information source) Because of the diversity of the selected databases, I would suggest that authors divide databases by types: bibliographic, complete text databases, etc. Because of the inclusion of medicine, social sciences, and bioethics, I would suggest including a few more databases covering all fields, such as Scopus and PubMed and related bibliographic databases. The authors should consider explaining or defining why did they take the 1980 year as a star year. The commencement dates of databases it is not relevant for the search strategy. For the protocol search, it would advise having the same start year. Regarding the grey literature, I would like to reassure the authors that they do not need to be afraid of the grey literature and consider using this protocol for application in the grey literature. To conclude, there are some crucial methodological changes, but if the suggestions are not applicable from the authors, I would like to receive more information to clarify the choice of the problems mentioned above.
--	---

REVIEWER	Borovečki, Ana University of Zagreb School of Medicine
REVIEW RETURNED	26-Jan-2022

GENERAL COMMENTS	Well written protocol applicable to the filed of research.
--

VERSION 1 – AUTHOR RESPONSE

Reviewer: 1

Dr. Dina Vrkić, University of Zagreb

Comments to the Author:

The authors of the proposed protocol are off to a good start; however, this protocol requires additional clarification and possible changes in the search strategy and information sources. Although it is a methodological issue, it should not disturb the flow of the proposed protocol.

Thank you to the reviewer for their comments.

In the search strategy, the authors are referring to the search strategy from the paper (ref. 18 - Schram A, Hougham G, Meltzer D, Ruhnke G. Palliative care in critical care settings: A systematic review of communication-based...) which is not so applicable for this protocol because it is off-topic. Therefore, the proposition is that the authors use mentioned article as an inspiration or starting point for key search terms in this protocol and explain why they used their selected MeSH terms and keywords.

While the systematic review by Schram and colleagues does not address the same research question as this systematic review, there is some overlap in the issues (end of life care) and setting (critical care). Hence, their search strategy was used as a starting point, but it was further defined through consultation with a librarian and review of the MeSH database. This has been explained in the manuscript.

The primary issue is a potential concept issue within the search strategy. Definition of the EOL decisions and relating them to “the withdrawal or withholding of life-sustaining treatments” conflict with the search term “Refusal to Treat” and focus on the aim of this protocol. Therefore, the authors should clarify the following section to avoid confusion. Please bear in mind that the “withdraw*” and “withhold*” keywords can connect to the “Withholding Treatment” MeSH term (<https://www.ncbi.nlm.nih.gov/mesh/68028761>), which is opposite of the “Refusal to Treat” MeSH term (<https://www.ncbi.nlm.nih.gov/mesh/68016079>) as is defined as: “Refusal of the health professional to initiate or continue treatment of a patient or group of patients”.

EOL decisions are here defined as the withdrawal or withholding of life-sustaining treatments, and in the context of ICU decision-making, the decision to withdraw/withhold could be the decision of the patient, the patient’s representatives, or the ICU team. The “Withholding Treatment” MeSH term relates to withholding treatment from a patient or research subject. This is differentiated from, but not necessarily the opposite of, “Refusal to Treat” – where the emphasis is on the health professional’s refusal to treat a patient, when the patient or patient’s representatives request treatment. Both of these scenarios can give rise to conflicts in EOL decision-making (although conflict is more implicit in the “Refusal to Treat” MeSH term), hence both of these MeSH terms have been included in the search strategy. Other terms, such as “Resuscitation Orders” have also been included by using the ‘Explode’ term. A footnote has been added to explain this - thank you to the reviewer for pointing this out.

Due to the wide range of years in the search, the search strategy needs to be more flexible to gather possible terms. For example, not all papers are indexed under the MeSH term “Terminal Care” from 1980 – until today. Therefore, the recommendation is that the authors need to include all possible MeSH and all possible keywords to the related term(s). In this specific case, it could also be helpful to use the MeSH term “Terminal Ill”, “Palliative Care”, etc. To be sure that all (possible) papers are retrieved.

Further MeSH terms and keywords to the related terms have been added to the search strategy, to ensure that all (possible) papers are retrieved. These have been evidenced in the supplementary file.

One suggestion, for the keyword, search the .mp search field is excellent in the Medline database for solo search, but because there is a multiple database search, I advise authors to

be consistent in the search and search fields across the other databases. Please consider to use .ab (abstract) and .ti (title) fields instead of the .mp.

Thank you to the reviewer for this suggestion. We have amended the search strategy to include “titles, abstracts, and keywords” as these are consistent fields across the included databases. Subject headings across the databases have still been utilised, where possible, and modified for the databases as needed. The full search strategy is documented in the supplementary file.

Also, please consider using the abbreviations of the EOL, (A)ICU in the search strategy because we never know how the title, abstract, or keywords are constructed. For example, authors can use (Ovid MEDLINE) for the “End of life”:

- (End adj2 life).ab,ti
- (End adj2 life care).ab,ti
- EOL.ab,ti.
- EOLC.ab,ti.
- EOL care.ab,ti.

Suggestions are on the same page as the definition and application of the methodological filters that the authors mentioned in the discussion.

Abbreviations have been included, many thanks to the reviewer for these suggestions.

The description of Boolean operators is redundant in describing the search strategy.

This has been removed.

(Information source) Because of the diversity of the selected databases, I would suggest that authors divide databases by types: bibliographic, complete text databases, etc.

The databases have been separated by type, as suggested.

Because of the inclusion of medicine, social sciences, and bioethics, I would suggest including a few more databases covering all fields, such as Scopus and PubMed and related bibliographic databases.

Scopus has been included. PubMed has not been included, as papers of relevance were not identified (when a sample of the records retrieved through PubMed were screened), beyond those already identified through MEDLINE.

The authors should consider explaining or defining why did they take the 1980 year as a star year. The commencement dates of databases it is not relevant for the search strategy. For the protocol search, it would advise having the same start year.

This has been amended to 2000, as critical care is a rapidly developing field, and studies older than 2000 were unlikely to be of relevance to current practice; however, we are aware of some papers from the early 2000s that fit the eligibility criteria.

Regarding the grey literature, I would like to reassure the authors that they do not need to be afraid of the grey literature and consider using this protocol for application in the grey literature.

While we recognise the potential value of including grey literature in systematic reviews, due to the time and resource constraints, we have been unable to include it. This systematic review is being undertaken at the outset of a PhD project, which will then go onto inform a qualitative study, and alongside a narrative review, in which some grey literature has been included. It would not, however,

be feasible to assess the full-texts of PhD thesis, for example, for eligibility. Similarly, conference abstracts or blogs may not offer enough detail. Therefore, grey literature has not been included.

To conclude, there are some crucial methodological changes, but if the suggestions are not applicable from the authors, I would like to receive more information to clarify the choice of the problems mentioned above.

Reviewer: 2

Dr. Ana Borovečki, University of Zagreb School of Medicine

Comments to the Author:

Well written protocol applicable to the field of research.

Thank you to the reviewer for their comments.